# A C-Type Lectin, RfCTL27, Activates the Immune Defense in the Red Palm Weevil *Rhynchophorus ferrugineus* (A.G. Olivier, 1791) (Coleoptera: Curculionidae: Dryophthorinae) by the Recognition of Gram-Negative Bacteria

**DOI:** 10.3390/insects15030212

**Published:** 2024-03-21

**Authors:** Yanru Gong, Yongjian Xia, Zhiping Su, Xinghong Wang, Yishuo Kou, Bing Ma, Youming Hou, Zhanghong Shi

**Affiliations:** 1State Key Laboratory of Ecological Pest Control for Fujian and Taiwan Crops, College of Plant Protection, Fuzhou 350002, China; gyr17852843301@163.com (Y.G.); xyj990819@163.com (Y.X.); m17820288537@163.com (Z.S.); 18285120070@163.com (X.W.); kys0511@163.com (Y.K.); ma15534605636@163.com (B.M.); ymhou@fafu.edu.cn (Y.H.); 2Ministerial and Provincial Joint Innovation Centre for Safety Production of Cross-Strait Crops, Fujian Agriculture and Forestry University, Fuzhou 350002, China

**Keywords:** invasive pest, pattern recognition receptor, insect immunity, antimicrobial peptide

## Abstract

**Simple Summary:**

Red palm weevil (RPW), *Rhynchophorus ferrugineus* (Olivier), is an invasive and highly destructive insect pest that poses a significant threat to palm trees. Currently, employing the biological agents are the important alternative method to fight against the infestation and alleviate the pesticide resistance of this pest. However, the control effect is unsatisfactory. It has been well known that the innate immunity plays the vital role for the protection of insect pests upon the attack of pathogens. In this research, we aimed to investigate the role of a C-type lectin, RfCTL27, in the immune response of RPWs. RT-qPCR showed that *Escherichia coli* induced a significant increase in the expression of *RfCTL27* in the gut and fat body. After *RfCTL27* was silenced, the expression level of antimicrobial peptide (AMP) genes in the gut and fat body was significantly decreased. Therefore, our findings indicate that RfCTL27 is involved in both systemic and gut local immunity by controlling the expression of AMP genes upon the exposure of Gram-negative bacteria.

**Abstract:**

Red palm weevil, *Rhynchophorus ferrugineus* (Olivier), is a palm tree insect pest that causes significant damage in the many countries from the Indian sub-continent and southeast Asia into date palm-growing countries of Africa, the Middle East, and the Mediterranean Basin. This study is aimed at determining the role of a C-type lectin, RfCTL27, in the immune defense of RPW larvae. RfCTL27 is a secreted protein that possesses a QPD motif, being integral for the discrimination of Gram-negative bacteria. The abundance of *RfCTL27* transcripts in the gut and fat body was significantly higher than that in other tissues. Six hours after injection of *Escherichia coli*, the expression level of *RfCTL27* in the gut of RPW larvae was significantly elevated compared with other groups. At 12 h after injection of *E. coli*, the expression of *RfCTL27* in fat body was dramatically induced in contrast with other treatments. More interestingly, the ability of RPW larvae to clear the pathogenic bacteria in the body cavity and gut was markedly impaired by the silencing of *RfCTL27*. Additionally, the expression levels of two antimicrobial peptide genes, *RfCecropin* in the gut and *RfDefensin* in fat body of RPW larvae, were significantly decreased. Taken together, these data suggested that RfCTL27 can recognize the Gram-negative bacterium and activate the expression of antimicrobial peptides to remove the invaded bacterial pathogens. This study provides a new scientific basis for improving the control efficiency of pathogenic microorganisms against red palm weevils in production practice.

## 1. Introduction

Red palm weevil (RPW), *Rhynchophorus ferrugineus* Olivier (Coleoptera: Curculionidae), is an alien destructive insect pest that seriously attacks palm trees and crops, such as sugarcane, and it has caused huge economic loss [1]. Currently, the primary strategy for controlling this pest is through the use of chemical pesticides. Unfortunately, increasing evidence has shown that *R. ferrugineus* has strong resistance to a variety of commonly used pesticides [2,3]. In the perspective of sustainable development, it is emergent to invent a novel alternative management strategy to fight against this pest. To date, certain biological agents, such as the entomopathogenic fungus *Beauveria bassiana* (Bals.-Criv.) Vuill. and bacterium *Serratia marcescens* J have been intensively found to have insecticidal activity against RPW larvae by bioassays in laboratory [4,5]. However, the control efficiency of these biological agents on *R. ferrugineus* in field is still out of our expectation [6]. It is well known that the innate immunity plays the vital role for the protection of insect pests upon the attack of pathogens. In this context, the elucidation of mechanisms by how insect pests activate their innate immunity to fight against the invaded pathogens is of great importance to improve the control efficiency of biological agents. The issuing of RPW genomic data [7] and the intensive usage of RNA-seq [5,8] advanced the identification of immunity-related genes of *R. ferrugineus*. Although some pattern recognition receptors (PRRs) have been determined by RNA silencing [9], the mechanisms underlying how the immune system of *R. ferrugineus* works upon the challenge of pathogens are still far from well known.

C-type lectins (CTLs) belong to a superfamily of proteins that are characterized by the presence of one or more C-type lectin-like domains (CTLDs, also known as carbohydrate-recognition domains, CRDs). These domains have the unique ability to recognize a variety of ligands and play a pivotal role in regulating animal immunity and maintaining homeostasis [10]. The CRD can specifically bind to galactose in the bacterial cell wall to recognize pathogens and activate immune responses in *Tribolium castaneum* [11]. Similar results have been reported in *Drosophila melanogaster* [12] and *Manduca sexta* [13]. These molecules can bind to carbohydrates in a Ca^2+^-dependent manner through conserved motifs, including EPN (Glu-Pro-Asn) and QPD (Gln-Pro-Asp), within the CTLD [13,14]. According to their structure, CTLs are usually classified into three families: the CTL-S family (single CRD), immulectin with dual-CRDs (the IML family), and the CTL-X family (many CRDs with other functional domains) [10]. It has been well verified that CTLs can function as PRRs to mediate animal immune responses, such as opsonization, nodule formation, encapsulation, melanization, and the production of antimicrobial peptides [10,15]. In this study, we combined RT-qPCR, RNAi, and the bioassays on the clearing ability of the invading pathogenic bacterium to validate the function of a CTL gene, *RfCTL27*, in the immune response of *R. ferrugineus*.

## 2. Materials and Methods

### 2.1. Red Palm Weevil Rearing and Breeding

Red palm weevil adults were bred in the incubator, and the parameters were set to a temperature of 27 ± 1 °C, photoperiod of 12L: 12D, and relative humidity of 75 ± 5%. Red palm weevil adults were bred in plastic boxes (7 cm in diameter, 10 cm high) and fed with fresh sugarcane chips, which were changed every 3 to 4 days. Female adults laid eggs in sugarcane chips. Then, the sugarcane chips were carefully dissected, and the eggs were collected with a brush and transferred to the Petri dish. Freshly hatched larvae were fed with fresh sugarcane chips. For this pest, the infestation of palm trees was mainly from the larval stage. Therefore, the healthy 4th instar larvae of RPW were selected for uncovering their immunity to the exposure of pathogenic microbes.

### 2.2. Sequence Characterization of RfCTL27

A C-type lectin (*RfCTL27*) was identified in the genome of the red palm weevil and validated by gene cloning. The Open Reading Frames (ORFs) finding tool (https://www.ncbi.nlm.nih.gov/orffinder/, accessed on 19 May 2023) and ExPASy-Translate tool (https://web.expasy.org/translate/, accessed on 19 May 2023) were used to predict the open reading frame and the encoded amino acid sequence of RfCTL27. To determine the presence of functional motifs and conservative active sites, we conducted multiple sequence alignments of RfCTL27 using MEGA-X and ClustalX 2.0. Then, we employed PhyloSuite 1.22 and IQtree software to construct phylogenetic trees of *RfCTL27* and other insect homologous proteins using the maximum likelihood method. Finally, FiguresTree 1.4.4 software was used to beautify the trees.

### 2.3. Analysis of RfCTL27 Expression in Different Tissues of the RPW Larvae

The 4th instar larvae of the RPW were dissected to obtain the head, cuticle, fat body, foregut, mid- and hindgut, and hemolymph. Three biological replicates were taken from each tissue, each containing three RPW 4th instar larvae. Then, total RNA was extracted from the above different tissues using Eastep^®^ Super Total RNA Extraction-LS1041. The concentration of total RNA was detected by NanoDrop 2000. RT-qPCR reverse transcription kit X0402 was used to prepare the template of cDNA. The reaction system was as follows: 4 μL 5× FasKing-RT SuperMix, 1 μg total RNA, and added RNase-free water to 20 μL. Primers for RT-qPCR were designed using Primer 3.0.

RT-qPCR was performed with Taq Pro Universal SYBR qPCR Master MixQ712-02. The RT-qPCRs were conducted with the following protocol: denaturation at 96 °C for 10 min, followed by 40 cycles of denaturation at 96 °C for 15 s, and annealing at 59 °C for 1 min. The RT-qPCR reaction system is shown in Table 1. *Rfβ-Actin* was used as an internal control in the experiment. The RT-qPCR data were calculated by 2^−ΔΔCt^ method.

### 2.4. Expression of RfCTL27 in the Gut and Fat Body of RPW Larvae Challenge with Pathogenic Microbes

To determine whether *RfCTL27* was involved in the immune response of RPW larvae against pathogenic microbe infection, the Gram-negative bacterium *Escherichia coli* E and Gram-positive bacterium *Staphylococcus aureus* Rosenbach were used to challenge RPW larvae by abdominal injection. *E. coli* and *S. aureus* were cultured overnight at 37 °C with OD_600_ = 0.65 in Luria–Bertani (LB) medium (10 g/L tryptone, 5 g/L yeast extract, 10 g/L NaCl, and pH 7.2). To prepare the bacterial suspension for challenging RPW larvae, 1 mL of bacterial solution was centrifuged at 5000× *g* for 8 min at 4 °C, and the supernatant was discarded. Then, the cell pellet was washed with 1 mL of sterilized PBS three times. Finally, the cell pellet was resuspended in PBS to make bacterial suspension with OD_600_ = 1.60. To challenge the RPW 4th instar larvae, 1 μL of bacterial suspension was injected into the body cavity. Totals of 6, 12, and 24 h after infection, the gut and fat body of these treated larvae were collected for total RNA extraction. Three biological replicates were set in each treatment, and each replicate contained four RPW larvae. RT-qPCR was employed to determine the expression level of *RfCTL27*, as described in Section 2.3.

### 2.5. The Impact of RfCTL27 Knockdown on RPW Larvae Immune Response and the Count of Gut eGFP-Labeled E. coli Colonies

Double-strand RNA (dsRNA) primers (Table 2) were designed using the E-RNAi (https://www.dkfz.de/signaling/e-rnai3/, accessed on 3 August 2023). *RfCTL27* and eGFP dsRNA were synthesized using MEGAscript^®^ Kit. eGFP dsRNA was used as a control. For gene silencing, 1 μL of dsRNA (1000 ng/μL) was injected into the fourth–fifth intersegmental membrane of the RPW 4th instar larvae. Three biological replicates were set, each containing four insects. Then 48 h of dsRNA injection, the gut and fat body were dissected.

To assess whether the silencing of *RfCTL27* impacts both systemic and gut-localized immunity in RPW, we used enhanced green fluorescent protein (eGFP)-labeled *E. coli* as the pathogen and attacked RPW larvae by abdomen injection and oral feeding. The eGFP-labeled *E. coli* was prepared for the following treatments according to the method described in Section 2.4. The 4th instar larvae of RPWs injected with dsRNA were fed for 45 h. Then, 1 μL of suspensions containing eGFP-labeled *E. coli* was injected into the body cavity of RPW larvae. A total of 3 h later, 100 μL hemolymph was extracted in a laminar flow cabinet. The hemolymph was placed in a sterile EP tube containing 3 μL alpha-phenylthiourea (PTU) solution (5 mmol/L) and diluted 1000 times in sterile PBS gradient. A total of 150 μL of hemolymph was distributed on LB solid agar plates that were fortified with ampicillin. The plates were placed upside down in a 37 °C incubator for 12 h, and then the colony-forming units (CFUs) of eGFP-labeled *E. coli* were observed under a microscope. The CFUs were carefully counted to ensure accuracy. For feeding treatments, we selected healthy 4th instar larvae for our study. These larvae were starved for 12 h after injecting ds*RfCTL27* and then fed with fresh sugarcane slices containing 1 mL of eGFP-labeled *E. coli* suspension. A total of 24 h later, the guts were taken out and ground into homogenate in 1 mL sterile PBS. The homogenate was diluted 1000 times, 100 μL of the gut homogenate was poured on LB mediums containing ampicillin. The counting method was the same as above. Three biological replicates were set in each treatment, and each replicate contained four RPW larvae. In addition, 45 h after *RfCTL27* silencing, the expression levels of the four antimicrobial peptide genes *RfColeoptericin*, *RfCecropin*, *RfDefensin*, and *RfAttacin* were analyzed after injection of *E. coli*.

### 2.6. Data Analysis and Processing

The RNAi efficiency and other experiments were detected by *t*-test. One-way analysis of variance (ANOVA) was used to detect the expression of *RfCTL27* in different tissues and the effect of bacterial injection on the expression level of *RfCTL27*. GraphPad Prism 8.0.1 software was employed to data analysis and drawing. The data in the figures are expressed in the form of mean ± SD.

## 3. Results

### 3.1. Sequence Characteristics of RfCTL27

The *RfCTL27* encoded a polypeptide, which is 217 residues long. The full length of *RfCTL27* was 1007 bp and ORF was 654 bp. *RfCTL27* contained two conserved functional domains, a carbohydrate recognition domain (CRD, 41–199 amino acids) and a signal peptide (1–17 amino acids), indicating that it belongs to the CTL-S family (Figure 1a). The binding specificity of *RfCTL27* mainly depends on the structural characteristics of CRDs. More interestingly, the Gln-Pro-Asp (QPD) motif was detected in RfCTL27, suggesting that it may recognize the pathogenic microorganisms, such as *E. coli*, by specifically binding mannose and galactose in their cell wall. Ca^2+^ binding sites were obtained by sequence alignment between *RfCTL27* and Rat MBP. The conserved Cysteine residues and two Ca^2+^ binding sites were discovered in CRD of RfCTL27 (Figure 1a,b) and calcium ions were required for the binding activity of *RfCTL27*. Phylogenetic analysis showed that RfCTL27 clustered with DmCTL-S4, DmCTL-S3, DmCTL-S2, and TcCTL-S2 (Figure 1c). This suggests that RfCTL27 is an orthologous protein in these species, indicating a shared evolutionary origin. Therefore, *RfCTL27* may have the same immune recognition function as CTL of *Drosophila melanogaster* Meigen and *Tribolium castaneum* (Herbst).

### 3.2. Expression Profiles of RfCTL27 in the Tissues of RPW Larvae

RT-qPCR assays showed that *RfCTL27* presented with significantly different expression levels in multiple tissues of RPW larvae, such as the head, cuticle, foregut, mid- and hindgut, fat body, and hemolymph (ANOVA: *F*_5,18_ = 4.24, *p* < 0.05). Notably, the expression level of *RfCTL27* was significantly higher in the gut and fat body than in the head, cuticle, and hemolymph (Figure 2), implying that *RfCTL27* might play some important roles in the local and systemic immunity of RPW larvae.

### 3.3. Expression Patterns of RfCTL27 upon Pathogen Challenge in RPW Larvae

After 6 h of *E. coli* injection, the expression level of *RfCTL27* in the gut of RPW larvae was significantly higher than that in other groups, i.e., 12 h and 24 h after *E. coli* challenge (ANOVA: *F*_2,6_ = 9.71, *p* = 0.013, Figure 3a). In fat body, the expression level of *RfCTL27* was significantly elevated at 12 h after the infection of *E. coli* in contrast with the phosphate-buffered saline (PBS) and *S. aureus* group (ANOVA: *F*_2,6_ = 43.36, *p* = 0.029, Figure 3b). Furthermore, the expression level of *RfCTL27* in fat body at 12 h after the challenge of *E. coli* was significantly higher than that of two other timepoints (ANOVA: *F*_2,6_ = 30.67, *p* = 0.043, Figure 3b). However, the injection of *S. aureus* into the hemocoel of RPW larvae did not increase the expression level of *RfCTL27* in fat body (ANOVA: *F*_2,6_ = 1.46, *p* = 0.31, Figure 3b). Totally, these results suggest that *E. coli* infection can affect the expression level of *RfCTL27* in the gut of RPWs, and *RfCTL27* in the fat body of RPWs may be involved in mediating the systemic immune response of the insect against Gram-negative bacteria.

### 3.4. Effect of RfCTL27 Silencing on the Immunocompetence of RPW Larvae and Expression of Antimicrobial Peptide Genes in the Gut and Fat Body

After 48 h of dsRNA delivery into the body cavity of RPW larvae, the expression levels of *RfCTL27* in the gut and fat body were significantly reduced. The expression level in the gut was significantly dropped by 96.20% (*t*-test: *t* = 7.12, *p* < 0.01), while that in the fat body was significantly dropped by 74.40% (*t*-test: *t* = 6.05, *p* < 0.01) (Figure 4). Moreover, a significant decrease in the relative expression level of the antimicrobial peptide *Rfcecropin* was detected in the gut of RPW larvae (*t*-test: *t* = 3.31, *p* < 0.05, Figure 5a). More interestingly, the number of eGFP-labeled *E. coli* colonies (CFU) was 12,500.00 ± 1946.79 in the gut of the fourth instar larvae of the RPW after RNAi interference with *RfCTL27*, while that of the control group was 7233.33 ± 550.76 (Figure 6). The number of the *dsRfCTL27* group was significantly higher than that of the *dseGFP* group (*t*-test: *t* = 4.51, *p* < 0.05), indicating that *RfCTL27* silencing significantly impaired the ability of the gut to eliminate the invaded pathogenic bacteria by oral feeding. These results suggested that RPW gut immunocompetence was compromised by the knockdown of *RfCTL27* to reduce the expression level of *RfCecropin*.

### 3.5. The Effect of RfCTL27 Knockdown on the Ability of RPW Larvae to Clear to the Invaded E. coli in Hemolymph

In the group of *RfCTL27* silencing, 16,146.67 ± 515.88 eGFP-labeled *E. coli* colonies (CFU) were recovered from the hemolymph of RPW larvae, but that of the control group was 7953.33 ± 853.31 (Figure 7). Statistical analysis revealed that the number of eGFP-labeled *E. coli* colonies in *RfCTL27*-silenced insects was significantly greater than that of controls (*t*-test: *t* = 14.23, *p* < 0.05). In fat body, the relative expression level of *RfDefensin* was also significantly decreased compared with the control group (*t*-test: *t* = 3.51, *p* < 0.05, Figure 5b), suggesting that the systemic immunity of this pest was impaired by *RfCTL27* silencing.

## 4. Discussion

Many studies have indicated an increase in *R. ferrugineus* immune resistance under the influence of infection with various microorganisms [1,8]. However, the mechanisms underlying how the immune system of RPWs detects and discriminates the invaded microbes have been mostly elusive. In invertebrates, CTLs have been shown to recognize bacteria and mediate immune responses [16,17]. Here, we found that *RfCTL27* encodes a polypeptide of 217 amino acids with two conserved functional domains, a CRD (41–199 amino acids) and a signal peptide (1–17 amino acids). The three-dimensional structure of RfCTL27 is a canonical fold, consisting of α-helices, β-sheets, and loops. This fold is stabilized by two pairs of disulfide bonds between Cys (71) and Cys (198), as well as Cys (171) and Cys (190). RT-qPCR analysis showed that the expression levels of *RfCTL27* in the gut and fat body of RPW larvae were significantly higher than that in other tissues. For insects, fat body and gut are the important immune organs, which are involved in the systemic and local immunity, respectively [18,19]. Consequently, the higher expression level of *RfCTL27* in these two tissues strongly suggest that *RfCTL27* might have vital immune functions. More specifically, our data indicated that the challenge of Gram-negative bacterium *E. coli* dramatically enhanced the expression of *RfCTL27* in the fat body and gut of RPW larvae compared with other treatments. The conservative functional domain, a QPD motif, was found in *RfCTL27*, which can specifically bind to galactose-type carbohydrates, peptidoglycan and lipopolysaccharide on the surface of bacterial wall [10,20,21,22,23,24]. Collectively, these results show that *RfCTL27* acts as a PRR to participate in the immune responses of this pest by the discrimination of Gram-negative bacterium.

After *RfCTL27* was silenced, we find that the ability of RPW larvae gut and hemolymph to clear eGFP-labeled *E. coli* was significantly reduced through biological experiments. Furthermore, we found that the silencing of *RfCTL27* led to a significant dropping in the expression level of two antimicrobial peptide genes, *RfCecropin* in the gut and *RfDefensin* in the fat body of RPW larvae. The antimicrobial peptides are the pivotal effectors for animals to clear the non-self objects including pathogenic bacteria and virus. Two antimicrobial peptides, Defensin and Cecropin, were identified in *Anopheles gambiae,* and Defensin was found to have antibacterial activity against *E. coli* [25,26,27]. In *Drosophila melanogaster*, the antimicrobial peptide Cecropin had stronger bactericidal activity against Gram-negative bacterium *E. coli* compared to Gram-positive bacterium *S. aureus* [28]. Consequently, these data evidently indicated that *RfCTL27* was concerned with the activation of AMP genes in this pest. The homeostasis of intestinal microbiota largely determines the healthy status of animal hosts [10,19]. CTLs modulate gut microbiome homeostasis by coating the bacterial surface to counteract AMP activity bacteria [29]. More specifically, the CTL24 protein prevents the continuous elimination of gut commensal bacteria, maintaining homeostasis by reducing LysC-mediated muropeptide release and AlfB1’s bacterial activity in shrimp [30]. In the current study, our results implied that *RfCTL27* could regulate gut immunity by the modulation of AMP expression. However, the mechanisms of how *RfCTL27* mediates the production of antimicrobial peptides and intestinal homeostasis are still unclear. In conclusion, our findings indicate that *RfCTL27* is related to the systemic and gut local immunity by regulating the expression of AMP genes under the exposure of Gram-negative bacteria.

## Figures and Tables

**Figure 1 insects-15-00212-f001:**
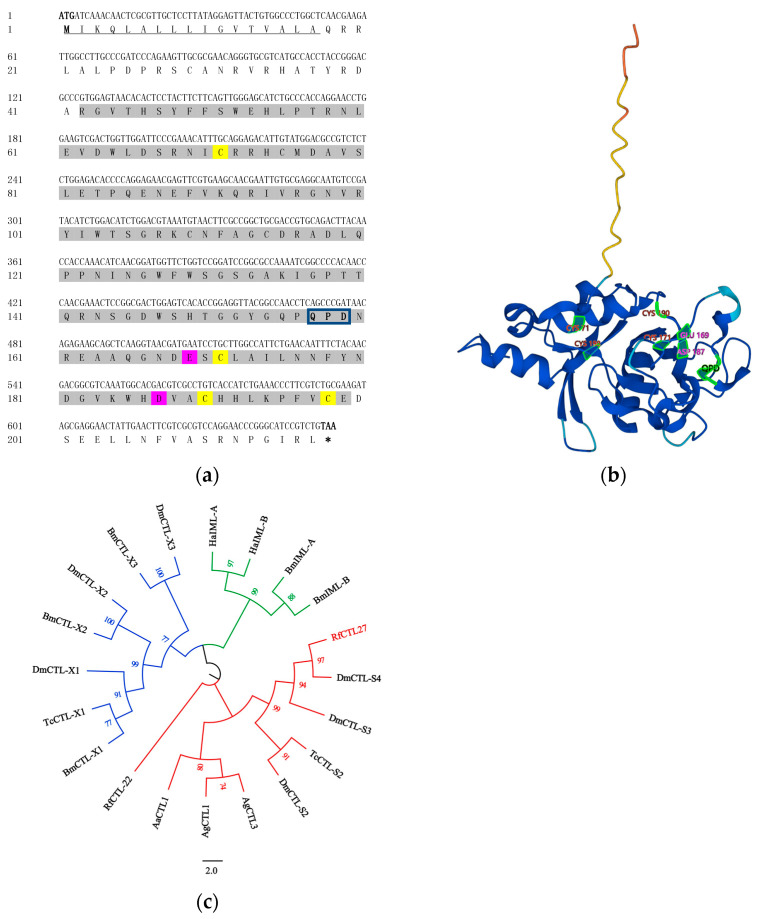
(**a**) The nucleotide and deduced amino acid sequences of RfCTL27. The black underline indicates the position of the signal peptide; gray shading indicates the location of the conserved domain CRD; blue boxes indicate the position of the conserved domain species QPD motif; yellow shading represents Cysteine residue sites; and red shading represents Ca^2+^ ions binding sites. (**b**) Protein tertiary structure of RfCTL27. Red represents Cysteine residue sites; purple represents Ca^2+^ ions binding sites; and black font stands for QPD motif. (**c**) Phylogenetic analysis of RfCTL27 with other insect CTLs. Branches are color-coded: red, blue, and green represent members of the CTL-S, CTL-X, and IML series, respectively. The Genbank accession number are as follows: *Aedes aegypti* AaCTL1 (XP_021694337.1); *Anopheles gambiae* AgCTL1 (AGAP004811-PA); *Anopheles gambiae* AgCTL3 (EAA13236.4); *Bombyx mori* BmCTL-X1 (XP004932809.1); *Bombyx mori* BmCTL-X2 (XP004932837.1); *Bombyx mori* BmCTL-X3 (XP021204064.1); *Bombyx mori* BmIML-A (NP001165368.1); *Bombyx mori* BmIML-B (NP001165396.1); *Drosophila melanogaster* DmCTL-S2 (NP_001260046.1); *Drosophila melanogaster* DmCTL-S3 (NP_650179.1); *Drosophila melanogaster* DmCTL-S4 (NP_001260199.1); *Drosophila melanogaster* DmCTL-X1 (NP001096984.1); *Drosophila melanogaster* DmCTL-X2 (NP001285146.1); *Drosophila melanogaster* DmCTL-X3 (NP001245864.1); *Helicoverpa armigera* HaIML-A (XP021189909.1); *Helicoverpa armigera* HaIML-B (XP021190921.1); *Rhynchophorus ferrugineus* RfCTL22 (KAF7282966.1); *Rhynchophorus ferrugineus* RfCTL27 (KAF7272888.1); *Tribolium castaneum* TcCTL-S2 (XP_624536.2); and *Tribolium castaneum* TcCTL-X1 (XP015833550.1).

**Figure 2 insects-15-00212-f002:**
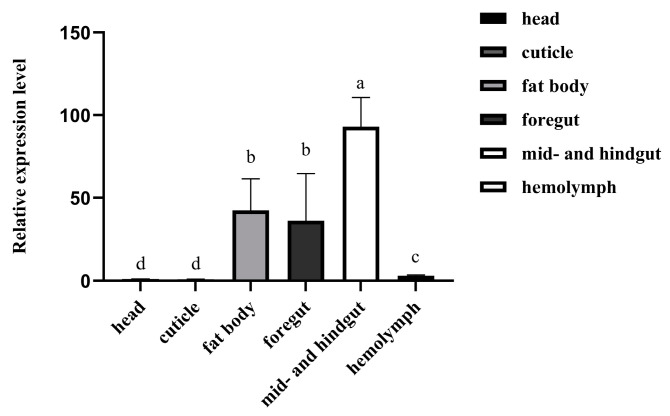
The expression profiles of *RfCTL27* in different tissues of RPW larvae. Four RPW larvae composed of a biological replicate, and three biological replicates were included in each treatment. *Rfβ-Actin* served as the internal reference gene. The data in the figure are represented by mean ± SD. Lowercase letters above bars indicate significance across different tissues (one-way analysis of variance (ANOVA) test, *p* < 0.05).

**Figure 3 insects-15-00212-f003:**
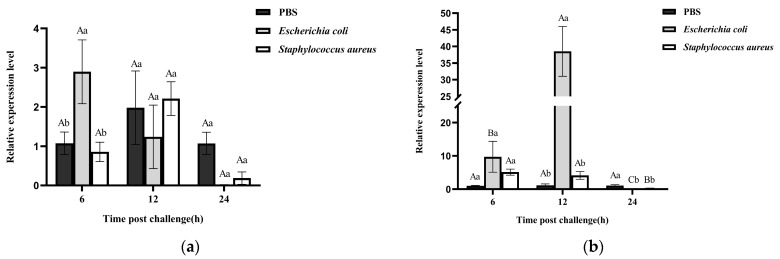
Changes in the expression level of *RfCTL27* in the (**a**) gut and (**b**) fat body of the RPW larvae upon challenge with *E. coli* and *S. aureus* by injection. Each larva was injected with 1 μL bacterial suspension, four RPW larvae composed of a biological replicate, and three biological replicates were included in each treatment. *Rfβ-Actin* served as the internal reference gene. Phosphate-buffered saline (PBS) was injected as a control. The data in the figure are represented by mean ± SD. Different capital and lowercase letters above bars represent significance in the gene expression level between different time points in the same treatments and across the different treatments at the same time point, respectively (one-way analysis of variance (ANOVA) test, *p* < 0.05).

**Figure 4 insects-15-00212-f004:**
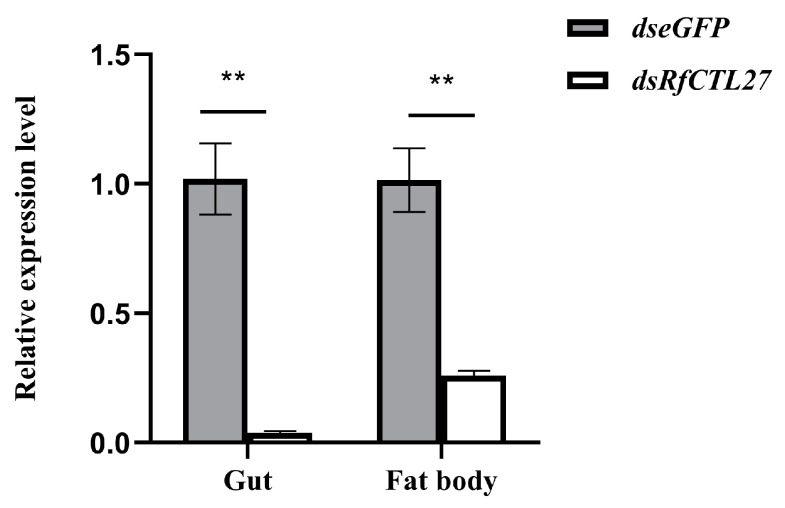
RNAi efficiency of *RfCTL27* in the gut and fat body of RPW larvae. *dsRfCTL27* represents the group for silencing *RfCTL27. dseGFP* served as the control by the delivery of double-strand *eGFP* RNA. Four RPW larvae composed of a biological replicate, and three biological replicates were included in each treatment. *Rfβ-Actin* served as the internal reference gene. The data in the figure are represented by mean ± SD. Asterisk above bars indicates significant difference in the figure between the two groups (*t*-test, *p* < 0.001).

**Figure 5 insects-15-00212-f005:**
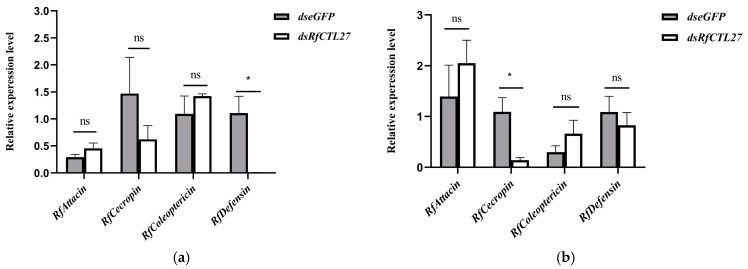
Effect of RNAi of *RfCTL27* on the expression levels of four AMP genes in the (**a**) gut and (**b**) fat body of RPW larvae. *dsRfCTL27* represents the group for silencing *RfCTL27. dseGFP* served as the control by the delivery of double-strand *eGFP* RNA. Four RPW larvae composed of a biological replicate, and three biological replicates were included in each treatment. *Rfβ-Actin* served as the internal reference gene. The data are represented by mean ± SD. ns represents no significance in the expression level of AMP genes between the two treatments. Asterisk above bars indicates significance between the two groups (*t*-test, *p* < 0.05).

**Figure 6 insects-15-00212-f006:**
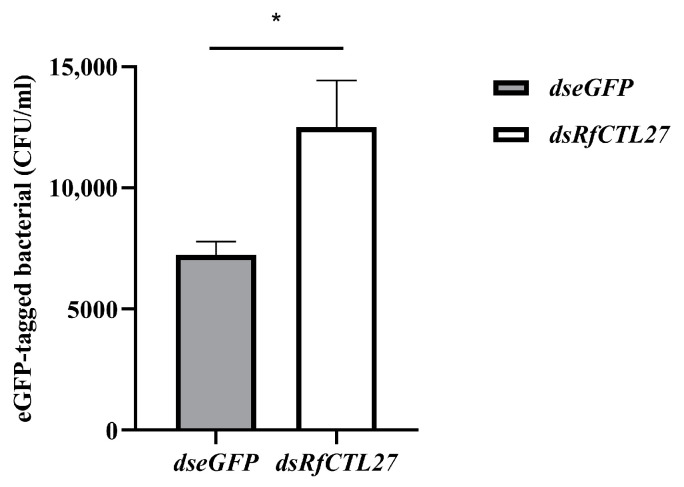
The effect of *RfCTL27* knockdown on the persistence of introduced eGFP-labeled *E. coli* in the gut of RPW larvae. *dsRfCTL27* represents the group for silencing *RfCTL27. dseGFP* served as the control by the delivery of double-strand *eGFP* RNA. Four RPW larvae composed of a biological replicate, and three biological replicates were included in each treatment. *Rfβ-Actin* served as the internal reference gene. The data in the figure are represented by mean ± SD. Asterisk above bars indicates significance in the figure between the two groups (*t*-test, *p* < 0.05).

**Figure 7 insects-15-00212-f007:**
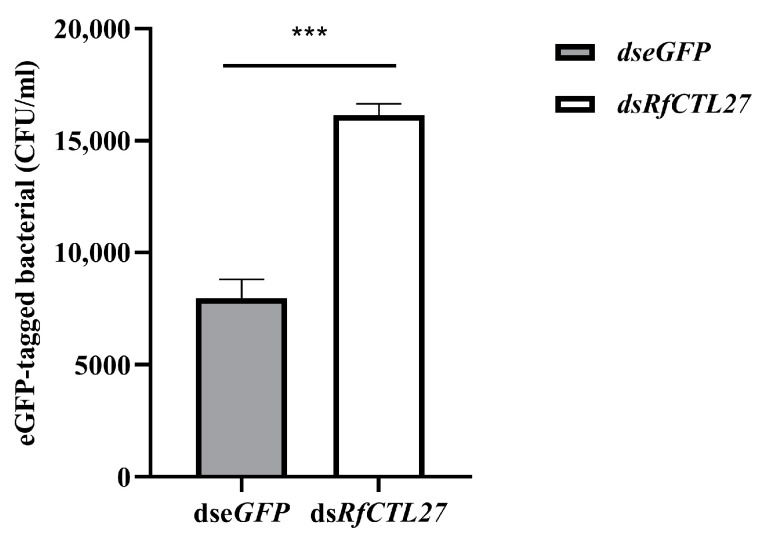
Effect of *RfCTL27* knockdown on the removing of injected eGFP-labeled *E. coli* of hemolymph in RPW larvae. *dsRfCTL27* represents the group for silencing *RfCTL27. dseGFP* served as the control by the delivery of double-strand *eGFP* RNA. Four RPW larvae composed of a biological replicate, and three biological replicates were included in each treatment. *Rfβ-Actin* served as the internal reference gene. The data in the figure are represented by mean ± SD. Asterisk above bars indicates significance between the two groups (*t*-test, *p* < 0.0001).

**Table 1 insects-15-00212-t001:** The composition of RT-qPCR reaction system.

Reagent	Dosage
SYBR mix	10 μL
Forward primer	0.4 μL
Reverse primer	0.4 μL
cDNA	1 μL
ddH_2_O	8.2 μL
Total	20 μL

**Table 2 insects-15-00212-t002:** The primers were used for RT-qPCR and RNAi.

Primers	Sequences (5′-3′)
RT-qPCR	
Rfβ-Actin F	CCAAGGGAGCCAAGCAATT
Rfβ-Actin R	CGCTGATGCCCCTATGTATGT
RfCTL27 F	ATCAACGGATGGTTCTGGTC
RfCTL27 R	ACGAAGGGTTTCAGATGGT
RfAttacin F	TGGTTCTGGTGCCCAAGTGA
RfAttacin R	GCCATAACGATTCTTGTTGGAGTA
RfCecropin F	CAGAAGCTGGTTGGTTGAAGA
RfCecropin R	GCAACACCGACATAACCCTGA
RfColeopericin F	TCGTGGTTTCTACCATGTTCACT
RfColeopericin R	TCAGCTAAAACCTGATCTTGGA
RfDefensin F	TTCGCCAAACTTATCCTCGTG
RfDefensin R	GGGTGCTTCGTTATCAACTTCC
RNAi	
RfdsCTL27 F	taatacgactcactatagggACCTGGAAGTCGACTGGTTG
RfdsCTL27 R	taatacgactcactatagggAGTTCCTCGCTATCTTCGCA
RfdseGFP F	taatacgactcactatagggCAGTGCTTCAGCCGCTAC
RfdseGFP R	taatacgactcactatagggGTTCACCTGCCGTTCTTGA

## Data Availability

The data presented in this study are available on request from the corresponding author.

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
