# Peer review of "A C-Type Lectin, RfCTL27, Activates the Immune Defense in the Red Palm Weevil Rhynchophorus ferrugineus (A.G. Olivier, 1791) (Coleoptera: Curculionidae: Dryophthorinae) by the Recognition of Gram-Negative Bacteria"

_insects, 2024, doi:10.3390/insects15030212_

Round 1

Reviewer 1 Report

Comments and Suggestions for Authors

Reviewer Comment and Recommendation for Manuscript ID: Insects-2886968-peer-review-v1

Title: A C-type lectin, RfCTL27, activates the immune defense in the red palm weevil Rhynchophorus ferrugineus Olivier (Coleoptera: Dryophthoridae) by the recognition of Gram-negative bacteria

General Comments:

In general, the manuscript effectively addresses the topic. Nevertheless, there is room for additional edits to enhance readability and conciseness. I suggest the authors consider incorporating my specific comments and suggestions to facilitate the acceptance and publication of this manuscript in MDPI-Insects

Specific comments and suggestions:

Abstract

 L.20 & L.27: Be consistent “RfCTL27, a C-type lectin” or “a C-type lectin, RfCTL27”

Keywords

L.40: “Rhynchophorus ferrugineus” and “C-type lectin” can be replaced; these words are already in the Title 

Introduction

L.43: Spell out “R. ferrugineus” and add the Order and Family names.

L.45: Change “[111]” 

L.56: Replace “invaed” with “invade”

L.73-75: The objective is not clear. Please, reword the objective (s).   

Materials and Methods

L.77: Spell out “RPW”

L.78: Spell out “RPW”

L.78-84: The authors should use punctuation marks appropriately to improve clarity

L.78 & L.79: Please, never start a sentence with an abbreviation (e.g. “RPW”); make corrections throughout the text.

L.81: Check and reword this “The sugarcane slices of RPW adults are .....”

L.83-84: Add a sentence in the text to explain why the 4th instar larvae of RPW used

L.86: Spell out “ORF” 

L.96: Change “Total” to “total”

L.131-139: The paragraph is not clear. Please, authors should apply all the necessary punctuations and reword the whole paragraph

L.145: “fourth instar larvae” & L.136 “4th instar larvae” be consistent 

L.145: Replace “48 h after dsRNA” with “After 48 h of dsRNA”; please, avoid starting a sentence with a number throughout the text

Results

L.192-197: Please move these lines to the Discussion Section

L.223; Add “in” after “roles”

L.226: “....in the figure” should be replaced with the referring treatments

L.229: What “other groups”? Please be specific

L.228-251: When listing the ANOVA parameters, for the p-value, write exactly as shown in the statistical output. Don’t just use P</> 0.05 when adding F and DF values

L.240: In Figure 3b, label the y-axis

L.260: “Figures 6” Remove “s” from “Figures”

L.268-269: “Asterisk above bars indicates significant ‘diference’ in the figure between the two groups” It looks more ambiguous. How many figures are the authors referring to here?

L.260-261: “former” and “later” replace these with the referring experimental units/treatments

L.257-260: Please write on “Figure 5” before interpreting “Figure 6”

 L.272-274: All the figures (1-7) captions need to be revised; they are very ambiguous

Discussion

L.315: Change ‘droppin” to “dropping”

L.294-329: The Discussion Section is brief compared to the available “Results”. Most of the discussions were done in the “Results Section” and these could have been moved to the “Discussion Section”. It is recommended that the authors merge the Results & Discussion Sections or move most of the lines in the Results Section to the Discussion Section and add little literature to the text. 

In general, the clarity of the whole manuscript was impeded by less efficient use of punctuation marks, wrong spellings, and other technical writing skills. However, this study has significant scientific merit. 

Comments on the Quality of English Language

In general, the clarity of the whole manuscript was impeded by less efficient use of punctuation marks, wrong spellings, and other technical writing skills. 

Reviewer 2 Report

Comments and Suggestions for Authors

In order for the publication to be of better quality, the manuscript should be corrected according to the reviewer's comments in the attached file "Reviewer`comments_REV1".

Reviewer 3 Report

Comments and Suggestions for Authors

This paper found that RfCTL27, a C-type lectin, was involved in immune defense of Rhynchophorus ferrugineus by controlling the expression of AMP genes upon the exposure of Gram-negative bacteria. The overall structure of the article is relatively complete and the amount of data is sufficient. However, there are still some problems that need to be revised before publication.

1.     Abstract: suggest to add the significance of the review at the end

2.     L36: “RfDefensin, in fat body” change to “RfDefensin in fat body”

3.     L45: “loss [111]” change to “loss [1]”

4.     L64-66: “These domains have the unique ability to…play a pivotal role in regulating animal immunity and maintaining homeostasis [12]”, suggest examples, especially those related to insects

5.     Section 2.2. “Sequence characterization of RfCTL27”, the sequence source of the gene is? Whether this study has done gene cloning, please briefly describe the experimental steps

6.     Section 2.3. “Analysis of RfCTL27 expression in different tissues of the RPW larvae”, how many fourth instar larvaes do these different tissues come from? How many repetitions are set up?

7.     L100: “1 μg total RNA, 1 μL Total RNA,”, please rewrite

8.     L134: “Then,The” change to “Then, the”

9.     L135-136: “10 μL of bacterial suspension was injected into the body cavity of the RPW 4th instar larvae.” What is the mortality rate of insects after injection? How many insects were injected into each group, and how many samples were taken

10.  Section 2.5. Please describe the injection details, including the concentration of injected dsRNA, the injection volume per larva, and the amount of injected larvaes

11.  L165-166: “Three larvaes were used as a biological replicate, and each replicate containing four fourth instar larvae.” please rewrite this sentence

12.  Figure 1: Please list the Latin full name of the species to which the used gene sequence belongs in the annotation

13.  L226: P < 0.05-“P” should be italicized, same as the entire text

14.  Figure 3-6: suggest to use the filling of the histogram to set the legend

15.  L255: “RfCTL27” change to “RfCTL27

16.  L257: “(t-test: t = 6.05, P < 0.01). (Figures 4)” change to “(t-test: t = 6.05, P < 0.01) (Figures 4)”

17.  Section 3.4. Suggest to change the order of Figure 5 and Figure 6

18.  L286: “In fat body, .” change to “In fat body, ”

19.  L287: “compared with the control group with the control group” change to “compared with the control group”

20.  L288-289: “suggesting that the systemic immunity of this pest was impaired by RfCTL27 silencing.” Whether this conclusion needs more experimental support or further discussion

21.  L315-318: Please discuss with more related literatures, do Cecropin and Defensin also play a role in the immune system of other insects

Round 2

Reviewer 3 Report

Comments and Suggestions for Authors

This revised version has addressed my comments.